# A Custom-Made Electronic Dynamometer for Evaluation of Peak Ankle Torque after COVID-19

**DOI:** 10.3390/s22052073

**Published:** 2022-03-07

**Authors:** Iulia Iovanca Dragoi, Florina Georgeta Popescu, Teodor Petrita, Florin Alexa, Romulus Fabian Tatu, Cosmina Ioana Bondor, Carmen Tatu, Frank L. Bowling, Neil D. Reeves, Mihai Ionac

**Affiliations:** 1Department of Vascular Surgery and Reconstructive Microsurgery, “Victor Babes” University of Medicine and Pharmacy, 2 Eftimie Murgu Square, 300041 Timisoara, Romania; contactfastfizioclinic@gmail.com (I.I.D.); frank.bowling@manchester.ac.uk (F.L.B.); mihai.ionac@umft.ro (M.I.); 2Discipline of Occupational Health, “Victor Babes” University of Medicine and Pharmacy, 2 Eftimie Murgu Square, 300041 Timisoara, Romania; 3Department of Communications, Politehnica University Timisoara, 2 Vasile Parvan, 300223 Timisoara, Romania; florin.alexa@upt.ro; 4Department of Orthopaedics-Traumatology-Urology and Imagistics, “Victor Babes” University of Medicine and Pharmacy, 2 Eftimie Murgu Square, 300041 Timisoara, Romania; tatu.fabian@umft.ro; 5Department of Medical Informatics and Biostatistics, University of Medicine and Pharmacy “Iuliu Hatieganu”, 8 Victor Babes, 400012 Cluj-Napoca, Romania; cbondor@umfcluj.ro; 6Department of Functional Sciences, Physiology, Center of Immuno-Physiology and Biotechnologies, “Victor Babes” University of Medicine and Pharmacy, 2 Eftimie Murgu Square, 300041 Timisoara, Romania; carmen.tatu@umft.ro; 7Department of Surgery & Translational Medicine, Faculty of Medical and Human Sciences, University of Manchester, Oxford Rd., Manchester M13 9PL, UK; 8Research Centre for Musculoskeletal Science & Sports Medicine, Department of Life Sciences, Faculty of Science and Engineering, Manchester Metropolitan University, Oxford Rd., Manchester M1 5GD, UK; n.reeves@mmu.ac.uk

**Keywords:** ankle joint, torque, muscle, skeletal, muscle strength, dynamometer, SARS-CoV-2

## Abstract

The negative effects of SARS-CoV-2 infection on the musculoskeletal system include symptoms of fatigue and sarcopenia. The aim of this study is to assess the impact of COVID-19 on foot muscle strength and evaluate the reproducibility of peak ankle torque measurements in time by using a custom-made electronic dynamometer. In this observational cohort study, we compare two groups of four participants, one exposed to COVID-19 throughout measurements and one unexposed. Peak ankle torque was measured using a portable custom-made electronic dynamometer. Ankle plantar flexor and dorsiflexor muscle strength was captured for both feet at different ankle angles prior and post COVID-19. Average peak torque demonstrated no significant statistical differences between initial and final moment for both groups (*p* = 0.945). An increase of 4.8%, *p* = 0.746 was obtained in the group with COVID-19 and a decrease of 1.3%, *p* = 0.953 was obtained in the group without COVID-19. Multivariate analysis demonstrated no significant differences between the two groups (*p* = 0.797). There was a very good test–retest reproducibility between the measurements in initial and final moments (ICC = 0.78, *p* < 0.001). In conclusion, peak torque variability is similar in both COVID-19 and non-COVID-19 groups and the custom-made electronic dynamometer is a reproducible method for repetitive ankle peak torque measurements.

## 1. Introduction

Sarcopenia is lately defined by the European Working Group on Sarcopenia in Older People (EWGSOP) as a muscle disease diagnosed by the presence of reduced muscle mass and strength or low muscle mass along with reduced physical performance [1]. Two-standard deviations below the sex-specific means were recommended for the diagnosis of sarcopenia in younger groups. In 2019, EWGSOP agreed on physical performance as a parameter used to categorise the severity of sarcopenia and not as a diagnostic tool. New cut-off points were also stated [2].

The risk to develop sarcopenia can be screened by patient self-reported questionnaires and a SARC-F questionnaire has been validated for this purpose in the elderly [3]. Probable sarcopenia is suspected when reduced muscle strength is diagnosed. The hand-grip strength (HGS) test is a valuable clinical tool that can be used to diagnose sarcopenia for both clinical and research purposes. In case HGS test cannot be used due to other conditions affecting upper limb and hand functional abilities, lower limb strength can be assessed [2]. Quantitative assessment of muscle strength using the HGS test can be performed with good applicability in research settings, specialist clinical settings and in primary care, while for the lower limb muscle strength, isokinetic/isometric tests could be performed [4]. Hand strength assessed with the HGS test demonstrated a good correlation with leg strength [5]. Hand-grip strength also correlated with lower extremity power, knee extensor torque and the calf muscle cross-sectional area [6,7].

Leg extension strength measured by isokinetic means needs both trained clinicians and adequate equipment [8]. When isokinetic means are not applicable, isometric torque measured with different methods and tools can be used to quantitatively assess lower limb muscle strength [9]. An alternative, but qualitative test used to assess lower limb strength and endurance, is the “chair rise test” previously cited by the EWGSOP. When the chair rise test was used instead of hand-grip strength, probable sarcopenia prevalence doubled across all ages [10].

For the evaluation of muscle mass, muscle quality and strength, imaging firstly and isokinetic/isometric means, respectively, are validated methods for the diagnosis of sarcopenia. In clinical practice, due to the equipment’s costs and clinicians training level, such tools could be underutilised. Lack of portability might be another aspect compromising clinicians’ resources to assess muscle parameters on site [4].

For this reason, clinical measurements and less expensive and more convenient tools are of need when muscle strength is required to be precisely measured. When no muscle mass diagnostic methods are available, calf circumference measurement can clinically predict performance and can be used in older people to approximate sarcopenia [11].

As sarcopenia has been attributed to all age groups, calf circumference might not be a valid tool to assess leg performance in the younger population. Analysing the strength of muscles acting around the ankle joint could also be an indicator of muscle performance, especially when gait efficiency is of clinical importance.

Portable dynamometers gained interest lately and papers have demonstrated even the custom-made electronic dynamometer’s reliability in measuring ankle torque [12].

Assessing maximal voluntary isometric contraction (MVIC) is a common method used to measure volitional maximal muscle strength, and normative data, despite being limited, were stated in previous studies. Reference values for lower limb MVIC were generated and could be used to facilitate the interpretation of measurements in relation to normal percentiles [13].

Muscle performance can be affected by age or by a multitude of conditions and even by inactivity [14] and derived sarcopenia can present in both acute and chronic forms [2].

COVID-19-related complications include symptoms that can persist and are long-lasting even after recovery. Persistent symptoms after the acute SARS-CoV-2 infection (“acute covid”), is a coined condition called “long covid” and it has been recognised in both mild and severe forms of COVID-19 [15]. COVID-19 classification based on the presence and severity of symptoms was published [16]. Acute and “long covid” effects on multiple organs and systems were stated, including the musculoskeletal system [17,18], with a negative impact on the skeletal muscle function even before the onset of other common symptoms [19]. Myalgia, atrophy and sarcopenia are part of the musculoskeletal manifestations’ spectrum [20]. Fatigue has also been reported as one of the symptoms encountered in both the acute phase and “long covid spectrum” [21]. Fatigue in relation to COVID-19 has been linked to general fatigue as a subjective complaint and was identified with subjective means on the Chalder Fatigue Scale (CFQ-11) [22].

It has been stated that SARS-CoV-2 infection might trigger a post-viral fatigue syndrome [23,24]. No association between COVID-19 severity and fatigue following COVID-19 was demonstrated. Highlighting the importance of screening for fatigue irrespective of severity of illness in patients recovering from COVID-19 is of interest [25]. The presence of the reported functional fatigue is defined as less strength in muscles on CFQ-11 and can be quantitatively assessed by reliable objective measurement devices, such as electronic dynamometers [12]. Linked to the musculoskeletal system, fatigue has been defined as any reduction in the capacity of a muscle to generate force [26] and can install whenever failure along the central or peripheral pathways results [27]. A decline in central activation can be defined as “central activation failure” or “central fatigue” [26]. Measurement techniques for the individual’s volitional abilities for maximal muscle activation were proposed [28].

Despite the decrease in the strength of respiratory muscles that has been documented [29], few data are available on specific muscle group strength prior and post SARS CoV-2 infection [30] and how this condition particularly influences muscle performance. Increased hand-grip muscle strength and a higher cross-sectional area of the vastus lateralis muscle on admission has been associated with a shorter hospitalisation period in patients with moderate to severe COVID-19 [31]. In the same severity categories of COVID-19, hand-grip suffered a reduction in strength when compared to controls and a reduction in distance over a six-minute walk was registered [32]. It is still unclear wheather central or peripheral paths are involved in muscle fatigue, in relation to SARS-CoV-2 infection.

Walking endurance could also be influenced by SARS-CoV-2 infection [15].

Foot involvement in gait performance is well known and implies both muscle strength and endurance, particularly during the contact phase of walking, when the ankle plantar flexors are considered to be the main participants [33]. Extensor muscles acting around the ankle joint are also involved in gait performance. Triceps surae, with the gastrocnemius muscle considered as the main ankle plantar flexor, produce internal plantar flexion moment around the ankle joint. Toes flexor muscles (flexor hallucis longus, flexor digitorum longus and the long toe flexor muscles) also function to plantarflex the foot; measuring the activity and strength of this particular muscle group is of relevance for understanding walking performance. Despite studies already demonstrating that hand grip suffered a reduction in strength in COVID-19 patients when compared to controls [32], a reduction in distance over a six-minute walk was registered [32] and walking endurance was influenced by SARS-CoV-2 infection [15], no particular study assessed whether SARS-CoV-2 infection directly affects foot muscles, as the key factor that further negatively influences gait parameters. In our opinion, COVID-19 could selectively affect certain muscle groups and not all skeletal muscles groups.

There is no data published on the small foot muscle function and strength assessed before contracting SARS-CoV-2 infection. Correlations between foot muscle fatigue/strength have not yet been reported in the literature; therefore, evaluation of the potential relationship between COVID-19 and ankle joint-related muscle weakness is the key innovation of our paper.

By assessing ankle torque, appreciation of foot muscle strength that act around the ankle joint in humans is possible [34]. Ankle torque can be precisely captured, but as per our knowledge, no previous studies on the evaluation of foot strength parameters measured at two different moments in time were reported when using a portable custom-made electronic dynamometer [12]. We did not find data published on ankle torque in relation to SARS-CoV-2 infection, the acute phase or “long covid”.

Our main objective was to assess the impact of COVID-19 on ankle torque, when measured at no longer than two months from the acute illness.

Our second objective was to evaluate the reproducibility of peak ankle torque measurements in time with a custom-made electronic dynamometer in a pilot study.

## 2. Materials and Methods

### 2.1. Participants

In this cohort study we reported data about 4 participants who, while participating in the non-intervention group of a trial study that included measurements of peak ankle torque at 2-month intervals, became positive for SARS-CoV-2 in November 2021. All 4 participants were confirmed with COVID-19 by positive nasopharyngeal ARN-Real Time PCR-STANDARD tests. This group was hereinafter referred to as the group 1 with COVID-19. The first set of measurements was performed on these participants as healthy subjects, before becoming infected with SARS-CoV-2, and the second set of measurements were performed during a second session after the imposed 14 days quarantine from the positive diagnosis.

Another 4 adult consenting participants, matched by gender from the same trial study non-interventional group, were randomly selected for this observational study as group 2 without COVID-19. These 4 participants did not report any symptoms of COVID-19 spectrum during the whole study period (including follow-up) and were not tested positive for SARS-CoV-2. Written and signed consent was obtained from all participants. Ethical approval from the University of Medicine and Pharmacy “Victor Babes” Timisoara Ethics Committee was released and registered under Nr. 50/21.09-14.10.2020. As participants in the non-experimental group of another study, any past/present unilateral/bilateral physical congenital or secondary deformities, abnormalities at the lower limbs, trauma/surgery, major neurological and cardiovascular conditions, or present psychiatric issues were considered exclusion criteria. Any adverse reactions, such as muscle pain, cramps or any physical/emotional discomfort during the measurements, were clearly presented to the participants and cessation of the measurement session was recommended in case such reactions might appear.

### 2.2. Disease Definitions

COVID-19 classification based on the presence of symptoms described five forms of severity [16]: asymptomatic or presymptomatic infection, mild illness, moderate illness, severe illness and critical illness.We used the latest definition for sarcopenia recommended by EWGSOP [2].

### 2.3. Data Collection

Anthropometric data, such as age, height and weight were registered/measured at the first visit.

The symptoms during the acute phase and persistent symptoms during the following weeks reported by the participants at the moment of measurements were registered.

Participants were asked during both sessions about any symptoms of sarcopenia with questions similar with those from SARC-F questionnaire. Any difficulties in walking in a room, transfers from chair/bed, climbing 10 stairs, unjustified falls in the past year and the associated reported symptoms of fatigue were registered.

During the first session data on the average number of daily steps of the last month were recovered from each participant wearable device application. Based on previously published normative data for healthy adults, a level of physical activity was established for each participant [35].

### 2.4. Description of the Measurement System and Measurement Procedure

#### 2.4.1. Measurement System Description

All participants had their ankle torque measurements assessed bilaterally. For ankle torque measurements, a reliable [12] portable custom-made dynamometer [36] was used during both separate occasions.

The dynamometer had the possibility to evaluate ankle torque at different ankle joint angles by pointing the apparatus pedal at various inclinations. Exact pedal angles were settled using an electronic inclinometer. The dynamometer construction permits the conversion of torque into force. The applied force on the load cell was further converted into voltage. The load cell is connected to a load cell amplifier that converts the load cell imbalance into voltage, further evaluated by oscilloscope means [37]. The personal computer (PC) connection with the oscilloscope permits through its PicoScope^®^6 software [38] the acquisition and recording of data for the whole measurement period. A full disclosure of the apparatus calibration and construction, measurement system and protocol intervention is described elsewhere [12]. Figure 1 represents the complete diagram of the measurement system components.

The oscilloscope software allows particular parameter selection of amplitude scale, time scale, resolution enhancement, etc. For the PicoScope graphic user interface the choice of parameters for configuration was: Channel A on, direct current (DC) coupling, input = 2 V and time/div = 100 ms/div (32 s length record), resolution enhancement 10 bits. An example of a flowchart with the setting steps guide for the selection of the parameters for the user graphic interface is presented in Figure 2.

The force cell provides a positive force value for active ankle plantar flexion and a negative value for dorsiflexion, transformed into voltage and further into Nm for torque. The captured voltage signal represented in V, as observed on the PC screen, corresponds to the variation of force value. When the participant’s foot is resting on the ground the obtained off-set voltage corresponds to the pedal’s own off-set level and this is represented in Figure 3—trace F. When the participant’s foot is relaxed on the pedal without the fixation strap being attached, a different off-set voltage can be observed in Figure 3—trace D. When the fixation strap is attached to the participant tight, just above the knee joint without being tightened, another off-set voltage can be observed in the Figure 3—trace C. When the foot is fixed in place on the pedal with the fixation belt tightened, but no participant voluntary action is placed on the dynamometer’s plate, a stable signal for the off-set level is obtained, as observed in Figure 3—trace E. A positive transition of signal from the off-set level for the force during participant’s active plantarflexion results as detailed in Figure 3—trace B. During active dorsiflexion a negative transition of signal from the off-set level results as shown in Figure 3—trace A. A complete representation for the six different outcomes was elaborated in MATLAB [39] and is represented in Figure 3.

Data recordings corresponding to each particular measurement were saved as folders from the PicoScope interface, corresponding to 32 text files containing voltage values. A MATLAB [39] application for the inspection of the saved text files was developed for further validation of the recordings. The application loads the multi-file contents of a measurement and concatenates them in a single time-graph. Inspection of the resulted time-graph permitted to the operator to further appreciate the quality of each resulted measurement and deciding whether the measurement fits for validation or another trial should be considered. Valid measurements were saved in folders for each participant with a numerical code and sent for data processing. Further low-pass filtering and scaling with pedal constant was applied, as previously described [12]. The data processor estimated relevant parameters of the performed measurements (voltage off-set and peak MVIC) and summarised the parameters into an Excel spreadsheet [40]. Excel data of voltage were later converted in torque data [12]. The data containing voltage values as well as the data converted into torque, could also be visualised as time-graphs. The resulted time-graphs as we have defined them are graphic representations in time of the voltage variations (expressed in V) as observed in Figure 4a, or torque variations (expressed in Nm) as observed in Figure 4b,c. On the Ox axis, the period is being represented in seconds (s) and voltage in V or torque in Nm on the Oy axis, respectively. On the torque time-graphs, peak torque values and off-set means are included. By software means for each recording we automatically computed the off-set mean (indicated by the red line in the time-graphs, as observed in Figure 4b,c) and the peak torque (highlighted with a red circle, as observed in Figure 4b,c).

#### 2.4.2. Methods of Participant Preparation for Measurement and Clinician Intervention

All participants had their ankle torque measurements assessed on two separate occasions. Previous to the SARS-CoV-2 infection, all 8 participants were scheduled at two months after the first measurement session. After knowing about the infection, SARS-CoV-2 positive participants were rescheduled, their second measurement being reprogramed in such a way that the participants were able to perform daily activities when they came for the second session measurements. SARS-CoV-2 non-diagnosed and asymptomatic participants had their ankle torque measured at baseline and after two months and were considered as a group without COVID-19.

Plantar-flexor and dorsiflexor muscle strength acting around the ankle joint have been captured. After the measurement protocol and with possible adverse effects fully detailed, participants were asked to sit on a chair with the knee and hip joints flexed and the examined foot plantigrade on the dynamometer plate, as shown in Figure 5a. The participants were asked to fully relax with their trunk resting on the chair backrest. For accurate participant positioning, nonelastic fibre belts were adjusted to ensure the fixation of foot and thigh. Tight fixation just above the knee joint allowed for the heel to remain fixed in place when ankle movements were performed. Another rigid strap was placed on the dorsum of the forefoot, just above the metatarsal-phalangeal joints (MPJ), allowing for a good fixation of the foot when analysing the dorsiflexor strength (Figure 5b). Placing the foot with the ankle joint axis of rotation (defined as the line passing through the medial and lateral ankle malleoli) right above the dynamometer’s pivotal point marked with a red line on the plate (Figure 5b), allowed for appropriate ankle torque measurements.

The participant position was selected based on previous research on the knee flexion angle in relation to the torque exerted by the gastrocnemius muscle. A decrement of maximal torque of the gastrocnemius muscle was observed when the knee joint was placed in a more flexed position [41]. By choosing this examination position, we were able to reduce the gastrocnemius impact as the primary ankle plantar flexor, allowing thereafter for a better isolation of the small muscles involved in plantar flexing the ankle (flexor digitorum longus, flexor hallucis longus and the long toe flexor muscles). Anterior tibialis muscle impact was reduced by placing the fixation strap on the dorsum of the foot just above the MPJ level.

Participants were instructed on the type of contractions. A succession of muscle efforts was requested and an experimental trial was initiated, without being considered a valuable measurement for analysis. As the participants fully understood the measurement procedure, voltage acquisitions started and clinician vocal commands to tense the muscles in order to flex the foot in the requested directions followed. After the experimental test, another trial of three MVIC was registered for both left and right ankle plantar flexor and dorsiflexor muscle groups. The duration of the acquisition was set to 32 s. This allowed for an efficient organisation of the succession of the three MVIC. Between the plantar flexor and dorsiflexor muscle efforts, for all pedal inclinations, participants had a two-minute recovery break in order to prevent muscle fatigue. Three MVIC of five seconds each were registered for left and right ankle dorsiflexor and plantar flexor muscle groups, leaving a five-second break between the three contraction sessions. The consecutive muscle efforts were registered at three pedal inclinations (0°, +5°, −5°), during active ankle plantar flexion and dorsiflexion, resulting in twelve recordings for each participant. When the tibia long axis is considered perpendicular to the ground parallel, we defined 0° of pedal inclination as the neutral ankle position (90° of ankle dorsiflexion), the +5° of pedal inclination as the 95° of ankle dorsiflexion and the −5° of pedal inclination as 85° of ankle dorsiflexion (which could also be referred to as 5° of ankle plantarflexion, starting from neutral).

All measurements were performed in the testing laboratory of the same physiotherapy unit placed in Timisoara, Romania between October 2021 and January 2022. The testing laboratory offered a constant room temperature (22 °C) and a generous space for an acclimatisation period of one hour before the trials. All measurements were recorded during a routine working day with a routinely physical activity level for each individual participant preceded by thirty minutes standing and walking in the same physiotherapy unit main building. Measurement recordings were later used for data processing.

To resume, we measured the passive moment at rest and ankle torque at 0°, +5° and −5° of pedal inclination during three consecutive MVIC of 5 s each separated by 5 s of relaxation periods (until the offset stability was gained), with the knee joint angle between 90° and 110° for both plantar and dorsiflexion, creating thereafter 12 measurements for each participant.

Results from the three consecutive MVIC, peak torque in Nm (described as the difference between the maximum obtained level of torque and the minimum obtained level of torque) and average values (between the peak torque resulted at the three inclinations) were registered and statistically analysed.

Peak values for torque were reported as representative values. We chose to not report the mean values of the three consecutive MVIC. Peak torques on the initial moment (baseline measurements) and on the final moment (second session measurements) were compared for the same individual participant.

Improper recorded data, both derived from human errors (participant and/or operator) or apparatus errors were eliminated.

#### 2.4.3. Methods for Validation of Acquired Data

The acquired data validation was made in three steps: one during the measurements and two during the interpretation of the results. During the measurements, errors due to participant indiscipline or improper clinician commands can appear. When the errors on the recorded voltage data were identified by the clinician, new trials were immediately requested.

Due to the oscilloscope limitations on the recorded data visualisation, not all the errors can be identified during the participant visit and collected data can still have errors. Recorded voltage from the oscilloscope as time-graphs were later inspected by the main researcher/operator and only valid measurements were selected.

Examples of improper acquisitions include participant errors (muscle efforts not corresponding with clinician vocal commands, improper stability of the off-set, improper contraction/breaks time in seconds and insufficient number of MVIC, etc.), as observed in Figure 6a, or testator command error, as observed in Figure 6b.

Elimination of improper recorded data, both derived from human errors (participant and/or operator) or apparatus errors was implemented by simple analysis of time-graphs obtained with the developed application.

An example of eliminated captured error due to the participant-reported pain during testation is represented in Figure 6c.

### 2.5. Statistical Analysis

Demographic data were compared between the two study groups with a *t*-test for quantitative normal distributed variables, Mann–Whitney test for quantitative non-normal distributed variables and Fisher’s exact test for qualitative variables. Normal distribution was tested with the Shapiro–Wilk test.

Time-graphs were analysed for the presence of errors. If errors were found, the corresponding peak torque was considered missing. In addition, some data might be missing due to fatigue, muscle pain or discomfort, which made the measurements impossible to be performed.

Clinical measurements were compared between the moments of measurements with the *t*-test for paired samples for normally distributed data and with the Wilcoxon signed rank test for non-normally distributed data. Clinical measurements were also compared between study groups with the *t*-test for independent groups in case of normal distributed variables or the Mann–Whitney test for non-normal distributed variables.

The % difference was computed with the formula 1 presented in the following:% difference = (Final_Moment − Initial_Moment)/Initial_Moment ∗ 100,(1)
where Initial_Moment was the first measurement of the peak torque and Final_Moment was the second measurement of the peak torque.

Clinical measurements in groups were checked for significant statistically differences between subjects, foot, flexion and ankle degrees with the Student’s *t*-test for independent samples in case of a variable with two categories and one-way ANOVA analysis with Tukey’s post hoc analysis in case of a variable with multiple categories.

A multivariate analysis of clinical measurements was made with linear mixed models.

The test–retest reproducibility between measurements in different moments was checked with the intraclass correlation coefficient (ICC) with a 95% confidence interval (CI). A statistical power analysis for the null hypothesis that “peak torque arithmetic means of the % difference between initial and final moment was not statistically significant different between Group 1 with COVID-19 and Group 2 without COVID-19” was performed [42]. *p* value was considered statistically significant for values smaller than 0.05. The analysis was made using SPSS application [43].

## 3. Results

### 3.1. Participant Characteristics

The participant demographic and anthropometric data were presented in Table 1. Group 1 with COVID-19 was comparable with group 2 without COVID-19 in terms of age, gender, body mass index (BMI) and foot length, with no statistically significant differences. Minimum age was 23 years old and maximum age was 52 years old. Minimum BMI was 21.36 kg/m^2^ and maximum BMI was 30.44 kg/m^2^. In the group 1 with COVID-19, one participant was considered somewhat active (~8000 daily steps) and three participants were considered definitely sedentary (~3500–~4000 daily steps). In the group 2 without COVID-19, two participants were considered active (~9000–~11,000 daily steps); the other two participants were sedentary (~1200–~3500 daily steps).

All four positive for SARS-CoV-2 participants (100%) were treated at home with no hospitalisation.

One participant reported particular symptoms during the acute phase, with burning type back pain with 8/10 on Visual Analogue Scale for pain, cough, mainly general fatigue and normal saturation. Fatigue persisted along with occasional back pain, paraesthesia in the left median nerve distribution area and subjective muscle weakness during left hand grip for the initial five weeks. At two months after recovery only general fatigue and dyspnoea during stair climbing was reported. A mild form of COVID-19 was attributed to this participant.

We agreed on a mild form of COVID-19 for three participants (75%), who reported headache (75%), fever (less than 38 °C, 75%), myalgia (75%), joint pain (50%), and general fatigue during the acute phase (75%) and persistent fatigue (50%) and subjective muscle wasting with stair climbing dyspnoea lasting for 8 weeks (75%). One participant reported associated respiratory function alteration during the acute phase (slightly reduced saturation, SpO_2_ ≥94%), which was considered a moderate form of illness.

On the first visit at baseline, no obvious functional impairments were self-reported by the enrolled participants (100%). On the second visit, only mild functional difficulties were self-reported when climbing stairs in three of four COVID-19 positive participants (75%) and fatigue in all participants (100%). The participants from group 2 without COVID-19 did not report any symptoms and/or functional difficulties.

The questions about sarcopenia symptoms had negative answers, with light fatigue being the only subjective symptom reported by all COVID-19 positive participants on the second visit.

### 3.2. Clinical Measurements Results and Statistical Analysis

Twenty-four measurements were made for each participant (six on each foot on each session, six on dorsiflexion and six on plantarflexion separately for each ankle inclination 0°, +5°, −5°).

For a clearer perspective, a representative time-graph was selected for each participant, including an overlap of the same type of measurement at the initial moment and final moment, as shown in Figure 7.

Reproducibility between the measurements was assessed by the intraclass correlation coefficient. There was a significant statistically very good reproducibility between the initial measurements and final measurements (both groups ICC = 0.78, 95% CI 0.67, 0.86, *p* < 0.001, group 1 with COVID-19 ICC = 0.68, 95% CI 0.41, 0.82, *p* < 0.001 and group 2 without COVID-19 ICC = 0.82, 95% CI 0.67, 0.90, *p* < 0.001).

There was a significant statistically good reproducibility between the initial measurements and final plantarflexion measurements (both groups ICC = 0.69, 95% CI 0.44, 0,84, *p* < 0.001) and for dorsiflexion measurements (both groups ICC = 0.84, 95% CI 0.71, 0.91, *p* < 0.001).

In Figure 8a, the arithmetic means of peak torque in group 1, with COVID-19 participants, were presented.

In Figure 8b, the arithmetic means of peak torque in group 2, without COVID-19 participants, were presented.

Errors were present in 6/192 (3.1%) measurements, four of them on the second participant (Figure 8a) in the final moment after COVID-19, two of them in the group 2 without COVID-19, one on the measurement in the first visit and the other in the second visit. The encountered errors were due to fatigue, muscle pain (Figure 6c) or discomfort in one of the measurements or due to impossibility to interpret the resulted time-graphs in another two of the measurements, due to indiscipline of the participant in four of the measurements (example in Figure 6a) and due to testator command error in the last one (Figure 6b).

The participant with missing data (second participant in Figure 8a) was measured at 3 weeks from the diagnosis of COVID-19. The other three participants in the analysis were measured at 10 weeks after the diagnosis (for two participants) and at 3 weeks after the diagnosis (participant no. 1 in Figure 8a).

In Table 2, arithmetic means (±standard deviation) of peak torque during MVIC for both groups, group 1 with COVID-19 and group 2 without COVID-19, were presented. In the case of group 1 with COVID-19, the average peak torque increased (but not statistically significant) with 4.81% between visits. In the case of group 2 without COVID-19, average peak torque decreased (but is not statistically significant) with 1.29% between visits.

The groups were comparable in the initial moment. When comparing the peak torque arithmetic mean of the % difference between the initial and final moment between the groups, no significant difference was found (*p* = 0.945 with a statistical power of 9%).

When the multivariate analysis was implemented on the group 1 with COVID-19, the results demonstrated no significant difference in peak torque between the flexion (Figure 9a), foot (Figure 9b) or angle (Figure 9c).

Multivariate analysis was implemented on the group 2 without COVID-19, the results being similar to the results presented for the first group. No statistically significant influence by different flexion (*p* = 0.649), foot (*p* = 0.147), or angle (*p* = 0.668) on peak torque was found.

When we introduced both groups in multivariate analysis, no statistical difference was found between them (*p* = 0.797) regarding % difference of the peak torque. We failed to demonstrate that COVID-19 had any influence on peak torque. These results confirmed the univariate analysis.

## 4. Discussion

Our study discloses that on our performed measurements SARS-CoV-2 infection had no negative impact on ankle torque assessed with our custom-made dynamometer. Our observations on the multiple MVIC reported the occurrence of a slight increment of ankle torque, with 4.81% when compared to baseline.

When measurements before COVID-19 diagnosis were compared with measurements after recovering from COVID-19, we did not find any significant reduction in ankle torque. Data were comparable with the unexposed group at COVID-19.

In one out of four participants, who had a mild form of COVID-19 (participant no. 2) we found a 44.5% decrease, measured after three weeks from the positive diagnosis. These impaired muscle results could not be explained by direct or indirect negative effects of COVID-19 on the foot skeletal muscles nor by the consequence of inactivity associated with the acute phase, as in the non-exposed to COVID-19 group we had two participants with near the same reduction of peak torque (30.43% and 18.42%, respectively).

SARS-CoV-2 infection has been demonstrated to impact the skeletal muscle’s functional outcomes. Associated symptoms (e.g., muscle pain, fatigue, muscle weakness, etc.) were also reported and a decrement of torque was observed for various muscles of the lower limb when isometrically tested [44]. During MVIC, a decrement was obtained when knee joint muscles were assessed in patients with COVID-19 [19].

Muscle strength reduction and sarcopenia was attributed as one of the SARS CoV-2 infection manifestations [45]. Other physio-pathological implications could also be involved in muscle wasting in patients assessed on hospital admission and after recovering from COVID-19.

Participants’ % difference between the initial and final moment was heterogeneous; for three participants, the ankle peak torque increased in group 1 with COVID-19, while for one participant, the peak torque decreased. In fact, the same heterogeneity was observed in the unexposed group, with decrement in the case of two participants and increment in the case of the other two participants.

At the first visit, no obvious functional impairments were self-reported by the participants. On the second visit, in three participants, only mild functional difficulties were self-reported when climbing stairs, lasting for one week after the acute phase. Light fatigue was the only subjective symptom reported by all participants at eight weeks after COVID-19 diagnosis. Despite light fatigue being self-reported by all participants in the group 1 with COVID-19, no associated muscle fatigue was demonstrated for the foot muscles when MVIC was measured by dynamometric means. This does not exclude fatigue of other muscle groups, as previously reported by other studies [30]. When we talk about the first objective of our study: to assess the impact of COVID-19 in our participants, one limitation was the small sample size, mainly due to the unplanned infection with SARS-CoV-2 of our participants. In this case, due to ethical issues, only an observational study can be done and not a planned experimental study. Sample size in such cases is the result of chance.

Our group was not representative for all forms of COVID-19. The characteristics of this disease divide the patients in multiple subcategories: patients with mild disease, moderate, severe disease and patients with critical illness [16]. Lack of significance between COVID-19 and non-COVID-19 groups might be explained by the absence of difference between peak torques, or by the reduced power of the study.

Further studies on a more statistical representative sample should be considered. Our portable custom-made electronic dynamometer could be an on-site device used in medical settings for measurement of reduced muscle strength. This could improve the sample size of captured ankle torque in the general population and further permit the follow-up of individuals later diagnosed with COVID-19 or any other unanticipated acute condition.

Our second objective in this study, when measuring ankle torque with a custom-made electronic dynamometer in different moments in time, was to evaluate the test–retest reproducibility between repetitive measurements. The proposed aim was achieved. We found a very good test–retest reproducibility between the repetitive measurements. As force variates in time [34], the longer the period between two measurements, the higher the variation. We demonstrated that the dynamometer used for testing had a good reproducibility of data over time, despite SAR-CoV-2 infection.

One study reported that the portable custom-made electronic dynamometer was a reliable method in measuring ankle torque in one session [12].

Reeves et al. used a similar device to the replica used in our study and repetitively assessed ankle torque in humans [46].

Isometric torque measured with different dynamometric methods and tools can be used to quantitatively assess lower limb muscle strength [9].

Maximal voluntary isometric contraction (MVIC) is a commonly used method for muscle strength measurements. In both healthy and affected by disease subjects, absolute maximum force alone may not be the only parameter used for the quantification of muscle performance [47]. In the future, measuring other parameters of muscle performance (e.g., endurance) with the given custom-made dynamometer could be taken in consideration. Analysing MVIC as a functional task, and not as a single maximal volitional effort, could open new insights in motor control levels.

When we talk about the second objective of our study: to assess the test–retest reproducibility of the dynamometer, a limitation of our work was the used small sample size. A sample size analysis was not possible for this pilot study due to lack of data in the literature, but a statistical power analysis was made and a small statistical power was found. In a future study, if we suppose that the % difference 6.1 will be maintained between the groups, the required sample size is found to be 65 participants in both samples [42]. Further studies are required on a greater number of participants, with each participant being measured two or more times in the same conditions. Subjects may change their force response in time due to exercise, lack of activity or other factors, and different physical/clinical conditions should also be taken into consideration in further studies.

Our study could be easily reproduced from the point of view of the second objective if one will dispose of a similar or the same portable custom-made dynamometer used, and the clinical measurements protocol would be rigorously respected. The obtained tendencies of this study will probably be observed when our study is replicated in the same conditions.

Further studies should consider custom-made dynamometry as a possible diagnostic method for COVID-19 or any acute condition-related reduced foot muscle strength.

## 5. Conclusions

In our study, the mild form of COVID-19 had no impact on ankle peak torque. We can comment on our results as being related to the small sample size and we consider that the same design with a bigger sample size in future studies could evaluate test–retest reproducibility with better precision. Another factor that contributed to the obtained results by limiting our sample size was the reduced probability in our group to develop COVID-19 while participating in another longitudinal study.

Peak torque variability in group 1 with COVID-19 was similar to the peak torque in group 2 without COVID-19. As peak torque during MVIC demonstrated no significant statistical differences between both groups, we believe future studies should also address other muscle parameters, such as endurance, force steadiness and force variation during repetitive MVIC, to better assess the possible implication of COVID-19 on foot muscle strength. Analysing MVIC as a functional task, and not as a single maximal volitional effort, could open new insights in motor control levels.

Our study demonstrated that the used portable custom-made electronic dynamometer is a very good reproducible method for repetitive ankle peak torque in dynamic.

We consider that further studies are needed on a greater number of participants with individual measurements repetitively recorded in two or more occasions in the same conditions.

## Figures and Tables

**Figure 1 sensors-22-02073-f001:**
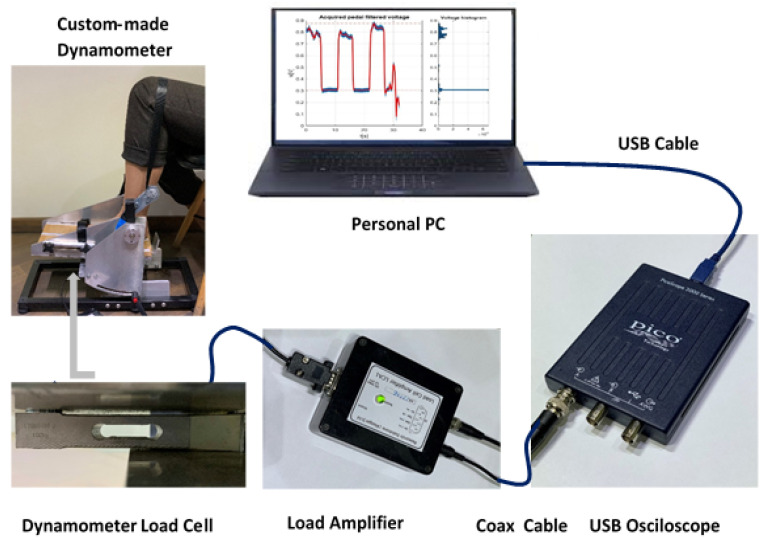
Block diagram components of the used measurement system.

**Figure 2 sensors-22-02073-f002:**
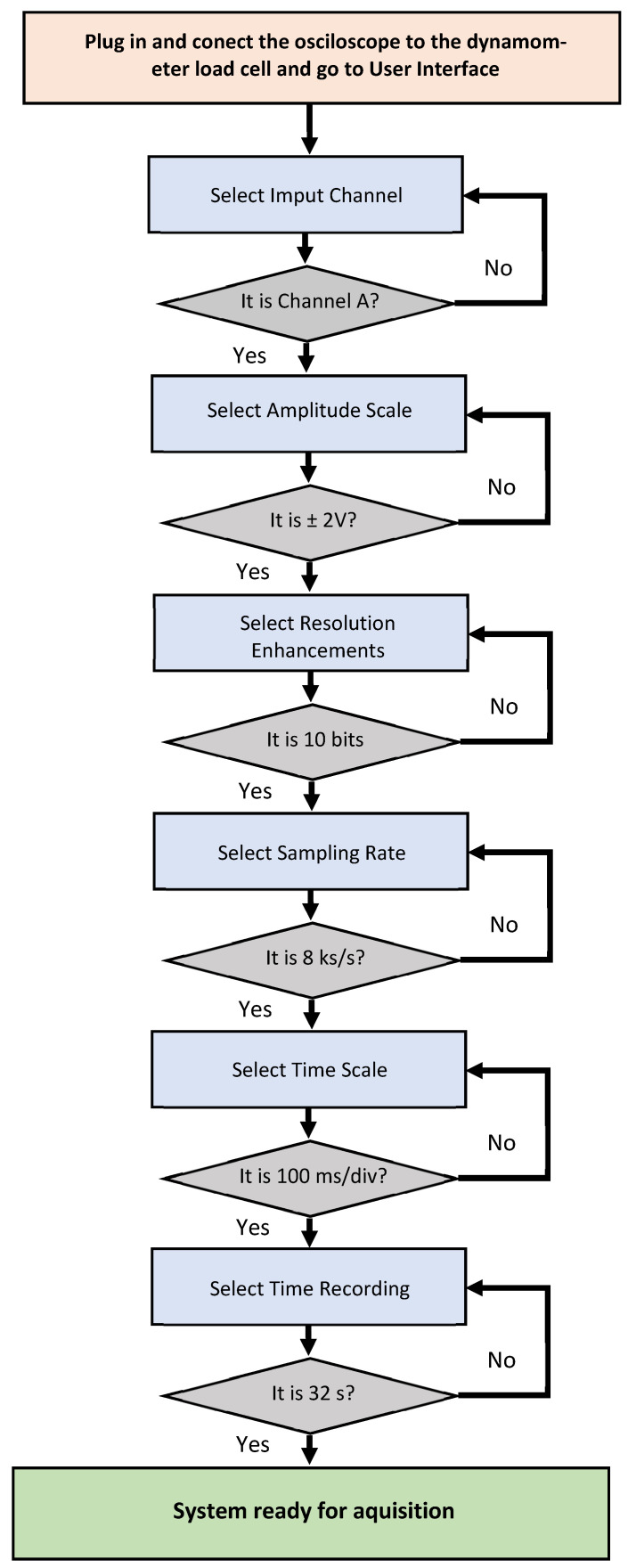
Flowchart for PicoScope6 parameters selection guide. Detailed graphic user interface and the selected parameters for the oscilloscope configuration: Channel A on, direct current (DC) coupling, input = 2 V and time/div = 100 ms/div (32 s length record) and 10 bits resolution enhancement.

**Figure 3 sensors-22-02073-f003:**
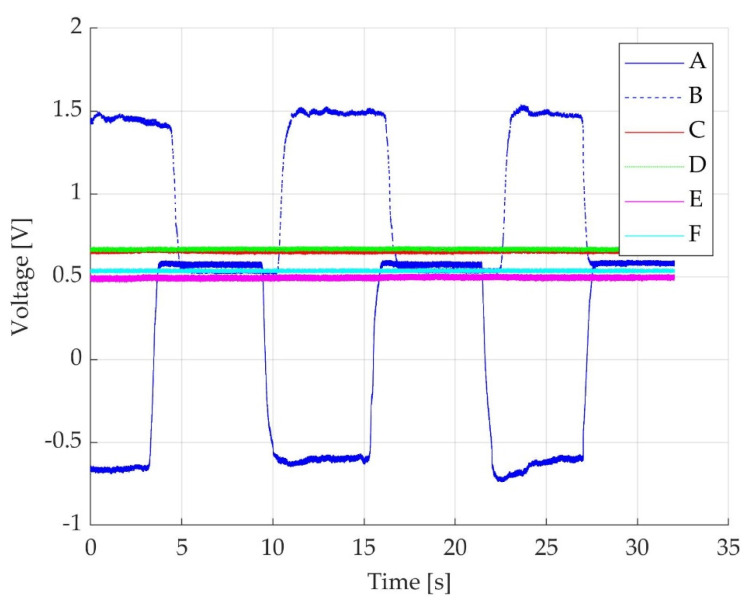
Voltage time-graph representation of six different outcomes: (trace **A**) an example of a negative displacement of voltage during three consecutive MVIC, while participant actively dorsiflexs the foot; pedal own mass voltage level, participant’s own limb mass voltage level and belt fixation-derived voltage levels are added to the voltage derived from the participant’s action placed on the pedal; (trace **B**) a similar example of a positive displacement of voltage, while participant voluntarily actions on the plate during three consecutive MVIC of active ankle plantarflexion; pedal own mass voltage level, participant’s own limb mass voltage level and belt fixation-derived voltage levels are added to the voltage derived from the participant’ action placed on the pedal; (trace **C**) off-set level with pedal own mass, participant’s own limb mass and un-tightened belt, without any voluntary action placed by the participant; (trace **D**) off-set level with pedal own mass, participant’s own limb mass without the presence of belt and no voluntary action placed by the participant; (trace **E**) off-set level with pedal own mass, participant’s own limb mass with tightened belt just above the knee joint, without any voluntary action placed by the participant; (trace **F**) pedal off-set level without the presence of the participant’s foot on the plate.

**Figure 4 sensors-22-02073-f004:**
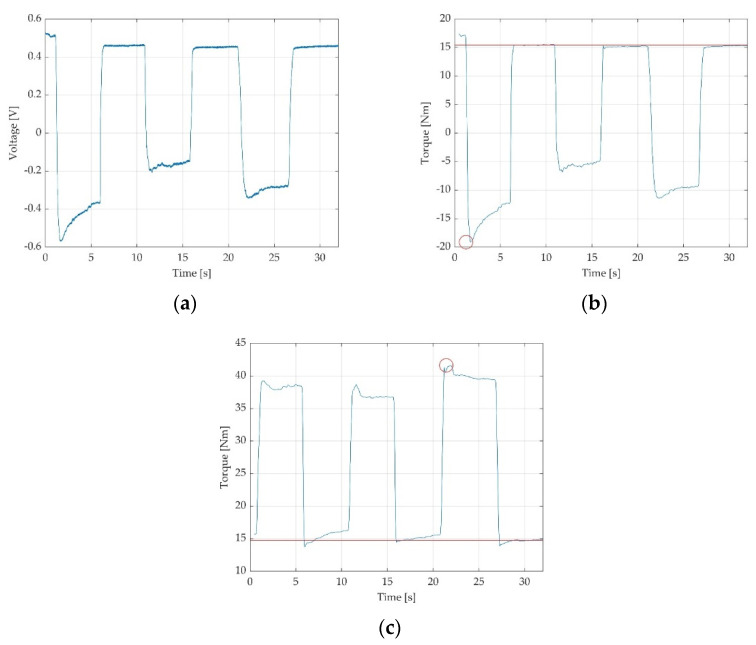
Three time-graphs of recordings, representing three MVIC of 5 s each followed by 5 s of relaxation between contractions: (**a**) resulted time graph of voltage in V during ankle dorsiflexion; (**b**) the same ankle dorsiflexion measurement represented as torque in Nm; (**c**) another example of valid time graph during ankle plantarflexion, representing torque in Nm.

**Figure 5 sensors-22-02073-f005:**
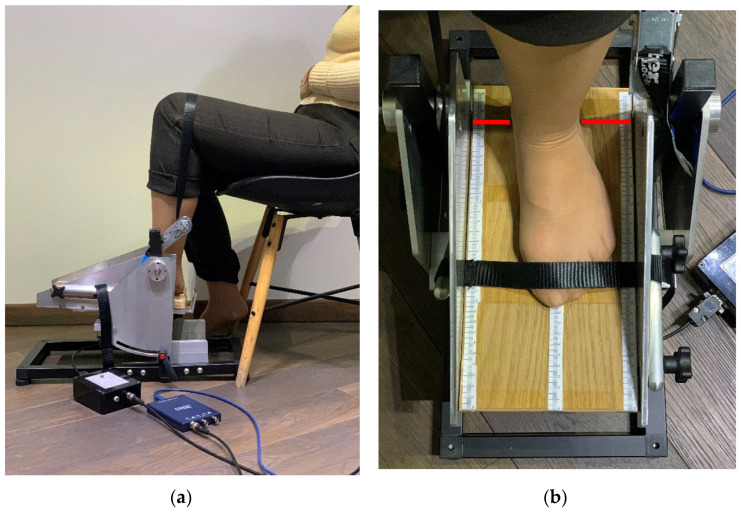
The participant’s position sat on a chair with trunk on the chair backrest: (**a**) fixation of strap just above the knee level, with tibia long axis perpendicular to the ground parallel and knee joint flexed at 90–110°; (**b**) strap fixation on the dorsum of the foot just above the MPJ joints, for a rigid fixation of the foot position on the dynamometer plate. The ankle malleoli axis is right above the pivotal line marked on the apparatus plate with a red line.

**Figure 6 sensors-22-02073-f006:**
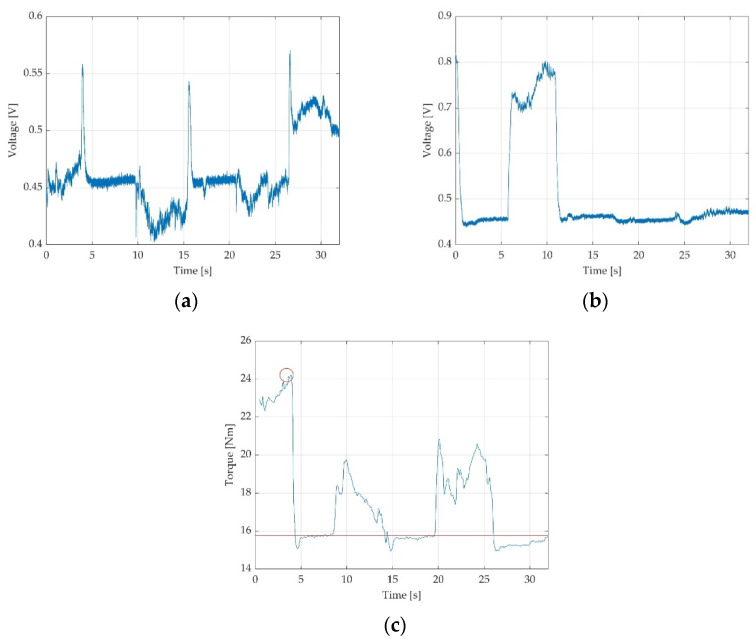
Examples of improper acquisition due to: (**a**) participant error; (**b**) testator command error; (**c**) an invalid time graph due to participants’ improper discipline/behaviour, mainly induced by pain during testation.

**Figure 7 sensors-22-02073-f007:**
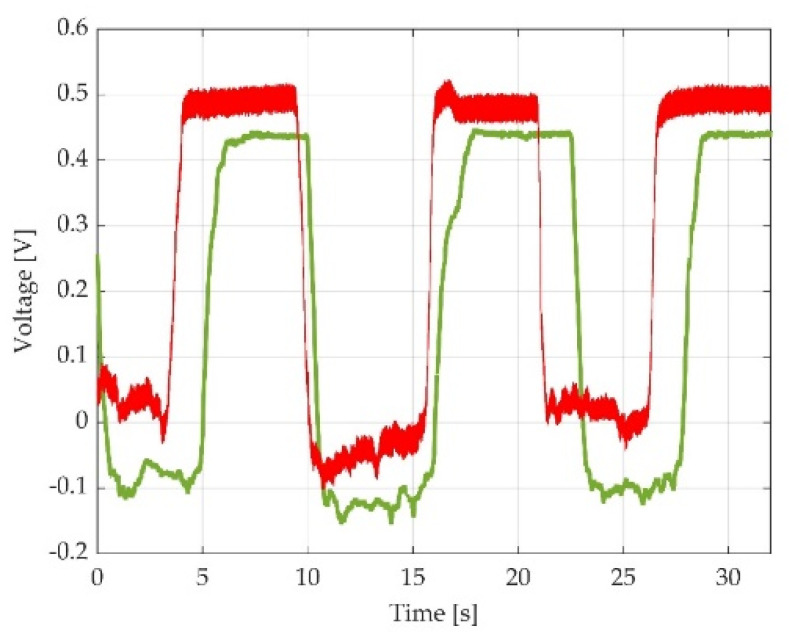
Representation of two overlapped voltage time-graphs selected for one participant during one type of muscle effort. Green line representing dorsiflexion at 0° of pedal inclination at initial moment; red line representing dorsiflexion at 0° of pedal inclination at final moment.

**Figure 8 sensors-22-02073-f008:**
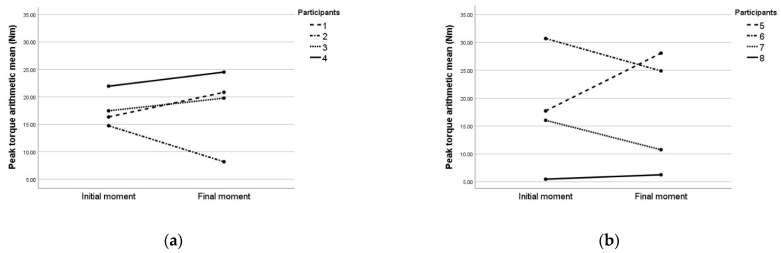
Arithmetic means of peak torque in each participant: (**a**) arithmetic means of peak torque in each participant from group 1 with COVID-19; (**b**) arithmetic means of peak torque in each participant from group 2 without COVID-19.

**Figure 9 sensors-22-02073-f009:**
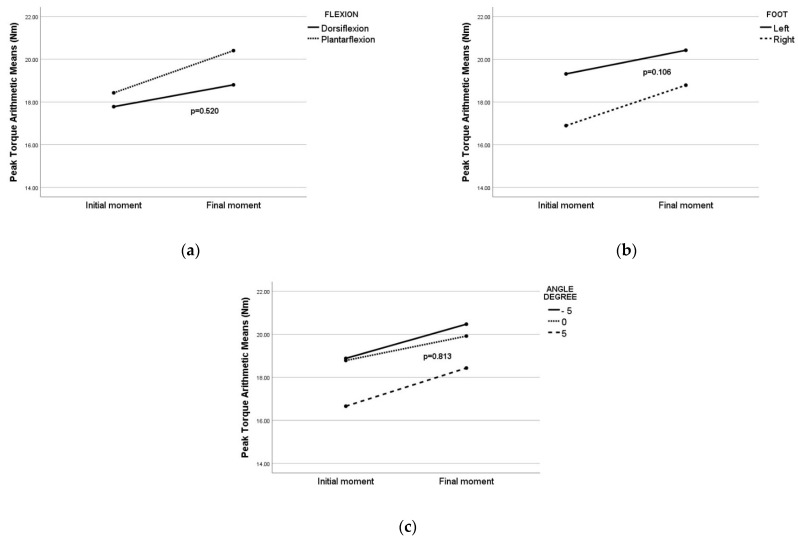
Arithmetic means of peak torque in group 1 with COVID-19: (**a**) arithmetic means of peak torque between flexion, *p*-value for flexion factor in multivariate analysis; (**b**) arithmetic means of peak torque between foot, *p*-value for foot factor in multivariate analysis; (**c**) arithmetic means of peak torque between angle, *p*-value for angle factor in multivariate analysis.

**Table 1 sensors-22-02073-t001:** The participant demographic and anthropometric characteristics in both groups.

Parameters	Group 1 with COVID-19 (*n* = 4)	Group 2 without COVID-19 (*n* = 4)	*p*
Age (years)	39.5 ± 14.15	31.5 ± 7	0.364 *
Male, no. (%)	2 (50)	2 (50)	1.000 **
Height (cm)	168.25 ± 12.42	170.75 ± 8.14	0.748 *
Weight (kg)	75.5 ± 11.96	66.75 ± 8.5	0.278 *
Foot length (cm)	25.05 ± 1.9	25.13 ± 2.21	0.961 *
Average no. of steps	4800 ± 2143.21	6175 ± 4588.66	0.614 *
BMI (kg/m^2^)	26.64 ± 3	22.82 ± 1.17	0.055 *
No. of weeks from diagnostics	6.5 ± 4.04	-	-

* *p* value obtained with *t*-test; ** *p* value obtain with Fisher’s exact test; no.—number; BMI—body mass index; arithmetic mean ± standard deviation.

**Table 2 sensors-22-02073-t002:** Mean peak torque (Nm) during MVIC for study groups in initial and final moment.

Groups	Initial Moment *	Final Moment *	% Difference **	*p*
Group 1 with COVID-19 (*n* = 4)	17.31 ± 3.51	18.14 ± 7.4	4.81	0.746
Group 2 without COVID-19 (*n* = 4)	17.53 ± 10.35	17.31 ± 10.39	−1.29	0.953
*p*	0.968 ^a^	0.900 ^b^	0.945 ^c^	

* Arithmetic mean ± standard deviation; ** % difference = (Final moment–Initial moment)/Initial moment*100; ^a^ *p*-value from comparison of the total peak torque in initial moment between group 1 with COVID-19 and group 2 without COVID-19; ^b^ *p*-value from comparison of the total peak torque in final moment between group 1 with COVID-19 and group 2 without COVID-19; ^c^ *p*-value from comparison of the % difference of increasing/decreasing peak torque between initial and final moment between group 1 with COVID-19 and group 2 without COVID-19.

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
