# Peer review of "A Custom-Made Electronic Dynamometer for Evaluation of Peak Ankle Torque after COVID-19"

_sensors, 2022, doi:10.3390/s22052073_

Round 1

Reviewer 1 Report

This paper presents research on peak ankle torque evaluation for covid-19 patients with custom-made electronic dynamometer. Average peak torque results showed no significant statistical differences between initial and final moment for both groups. The following comments should be considered before publication.

  1. The Introduction part should be reconstructed for better understanding. I did not find out the potential relationship between covid-19 and ankle joint related muscle weakness, which is the key innovation of this paper. The authors should point out this issue clearly in this section.
  2. The figures in this manuscript is quite low in this version. I do not think it meets the standardization of the journalSensors. Please replace figure 2 and figure 3 with clear and scientific figures rather than screen captures. Maybe figure 2 can be replaced by flow chart that shows your UI or steps of parameters configuration.
  3. The Conclusion section just simply summarized the results again. Maybe the authors can briefly discuss why this is no significant difference, how to further design the experiment to make the results more reliable.

Author Response

Dear Reviewer, 

The Authors team appreciated your efforts and time to review our work.

We hope that we have managed to answer point to point to your pertinent comments and kind suggestions.

Thank you very much. We are waiting for your answer. 

All the best!

Florina Popescu and the Authors

Reviewer 2 Report

The Authors introduced a custom measurement system for evaluating the effect of COVID-19 on lower limb muscle performances. They did not find any significant evidence of such effects, thus concluding that COVID-19 had no impact on ankle peak torque.
The authors carried out a significant amount of work. However, there are some critical flaws.

1. The lack of differences between COVID-19 and non-COVID-19 subjects is likely to be due to a lack of statistical power. A power analysis should have been taken into account, based on previous evidence, even if previous studies mainly focused on acute illness cases. The Authors only hinted at such limitation.

2. Regarding the reliability of the instrument, the Authors assessed it through correlation between initial and final (~after 2 months) measurements. This approach may be misleading, as subjects may change their force response due to excercise, lack of activity or other factors. It would be better to assess repeatability between measurements taken from a great number of subjects 
with different physical/clinical conditions, measuring each subject two or more times in the same conditions. By doing so, even with the data they have already acquired, they would properly characterize the precision of the instrument, both at the initial and final moment.

3. In order to compare measurements from two classes of subjects (e.g. COVID19 and non-COVID19), taken in two or more different moments
 (e.g. baseline and after two months), a mixed-model analysis should be performed.

A few more observations:
- when comparing paired samples, the Authors refer to Wilcoxon test without any other specification: they should refer to "Wilcoxon signed rank test"
- English check is needed.

Author Response

Dear Reviewer, 

The Authors team appreciated your efforts and time to review our work.

Hoping that we answered point to point to all your pertinent remarks and kind suggestions, we thank you very much and we are waiting for your answer.

All the best, 

Florina Popescu and the Authors

Round 2

Reviewer 2 Report

I have no more comments. Just do a last spell-check (e.g.: line 404, better using "chose" for tense consistency).

Author Response

Dear Reviewer, 

Thank you for your pertinent observations. We hope we have all English language and style, fine/minor spell check edited.

Line 404 was corrected. 

Thank you very much for your kind support.

Hoping we have reached  journal expectations, 

The Authors